# LLaVA-Read: Enhancing Reading Ability of Multimodal Large Language Models

## Abstract

Multimodal large language models have demonstrated impressive capabilities in understanding and manipulating images. However, many of these models struggle with comprehending intensive textual contents embedded within the images, primarily due to the limited text recognition and layout understanding ability. To understand the sources of these limitations, we perform an exploratory analysis showing the drawbacks of classical visual encoders on visual text understanding. Hence, we present LLaVA-Read, a multimodal large language model that utilizes dual visual encoders along with a visual text encoder. Our model surpasses existing state-of-the-art models in various text-rich image understanding tasks, showcasing enhanced comprehension of textual content within images. Together, our research suggests visual text understanding remains an open challenge and an *efficient* visual text encoder is crucial for future successful multimodal systems.

## 1 Introduction

Instruction tuning (Ouyang et al., 2022; Chung et al., 2022) has demonstrated remarkable generalization abilities across unseen tasks, contributing to the increasing adoption of large language models (LLMs) such as GPT-4 (OpenAI, 2023). Recently, multimodal language models have benefitted from visual instruction fine-tuning (Liu et al., 2023c; Li et al., 2023a; Li, 2023; Zhu et al., 2023; Alayrac et al., 2022), leading to significant successes in real-world applications. These models utilize visual encoders such as CLIP-ViT (Dosovitskiy et al., 2020; Radford et al., 2021) to imbue LLMs with image comprehension capabilities. However, challenges persist in comprehending textual information within images, likely stemming from the prevalence of natural images in training datasets such as Conceptual Captions (Changpinyo et al., 2021) and COCO (Lin et al., 2015)), as highlighted by (Liu et al., 2023e). To address this, (Zhang et al., 2023d) proposed improving end-to-end visual instruction-tuned models by introducing noisy Optical Character Recognition (OCR) annotations to improve vision language alignment. Additionally, low-resolution visual encoders pose challenges as a minimum of nine pixels are required to recognize a word. Previous works (Liu et al., 2024b; Bai et al., 2023; Dong et al., 2024) have explored various methods to improve encoder resolution, resulting in significant performance gains in various downstream tasks. However, it is worth noting that high-resolution encoders typically require more resources for image encoding and produce more visual tokens for language models to process, leading to inefficiencies in training and inference. (Li et al., 2023d; Chu et al., 2023) have proposed methods such as visual token merging and smarter architecture designs to mitigate these challenges and enhance model performance.

Document images often comprise text-rich content, with the visual components typically being simple while the textual parts are densely packed. A pertinent inquiry arises regarding the proficiency of existing visual encoders in encoding visual text and generating visual tokens for language models. To address this, we conducted synthetic experiments to assess visual encoders' performance in text recognition and compare it with open-source Optical Character Recognition (OCR) tools. Our analyses reveal that OCR tools exhibit superior efficiency and accuracy in encoding large text blocks, whereas popular visual encoders excel in recognizing smaller and shorter words and phrases. In addition, OCR tools can seamlessly scale up to process high-resolution images at minimal cost. Motivated by these findings, we propose a novel architecture named LLaVA-Read that integrates multiple visual encoders. Our rationale dictates that a visual encoder should efficiently capture visual information, while a lightweight visual-text encoder (e.g., OCR tools) extracts text from high-resolution images. Furthermore, we explore the integration of a high-resolution visual encoder

into LLaVA-Read without increasing the number of visual tokens for language models, achieved through a fusion module. To enhance alignment and collaboration among dual visual encoders, we leverage both text and layout information from visual-text encoders, introducing various layout-aware pretraining and fine-tuning tasks. These efforts yield significant improvements in the understanding of text-rich images. In summary, our contributions are threefold:

- We conduct a comprehensive analysis of the text recognition capabilities of multimodal large language models, which reveals their impressive capability on scene text understanding but limited proficiency in comprehending large amounts of textual content within a text-rich image.
- We propose LLaVA-Read, a model architecture adept at efficiently encoding textual and visual information. The use of multiple visual encoders, including a lightweight visual-text encoder, enables efficient extraction of visual texts.
- LLaVA-Read, coupled with layout-aware pretraining and instruction finetuning, demonstrates substantial enhancements in text-rich image understanding, surpassing multiple baselines on public benchmarks.

## 2 RELATED WORK

**Multimodal Instruction Tuning**   Multi-modal instruction tuning, including image (Liu et al., 2023c; Dai et al., 2023; Alayrac et al., 2022), video (Zhang et al., 2023b; Maaz et al., 2023), and audio (Huang et al., 2023; Zhang et al., 2023a) settings, has been an active research topic. Most efforts aim to integrate visual representations, which are obtained through an independent visual encoder, into large language models. MiniGPT-4 (Zhu et al., 2023) uses ChatGPT to generate high-quality instruction-following data, while LLaVA (Liu et al., 2023c) generates such data by prompting GPT-4 with captions and bounding boxes. Previous works  (Chen et al., 2023; 2024) generate more than 1M high-quality data for multimodal LLM training via prompting OpenAI GPT-4V. LLaMA-Adapter (Zhang et al., 2023c; Gao et al., 2023) aligns text-image features using COCO data, and mPLUG-owl (Ye et al., 2023b) combines extensive image-text pairs for pretraining and a mixture of data for fine-tuning. InstructBLIP (Dai et al., 2023) addresses this by transforming 13 vision language tasks into an instruction-following format.  mPLUG-Owl (Ye et al., 2023a;b) apply multitask instruction funetuing using existing document datasets. Previous works (Liu et al., 2023b; 2024b; Bai et al., 2023; Dong et al., 2024; Xu et al., 2024; Luo et al., 2024) have investigated different ways to improve encoder resolution, receiving great improvement in various downstream tasks. A comprehensive survey is available (Li et al., 2023b). Despite this, many models struggle with visual text understanding tasks (Liu et al., 2023e). The proposed LLaVA-Read aims to improve the text-rich image understanding ability, where both visual objects and visual texts understanding can be done simultaneously.

**Visual Document Understanding**   There have been efforts to boost multimodal LLMs to better comprehend text-rich images, including document images. Among these, LLaVAR (Zhang et al., 2023d) uses GPT-4 to collect fine-tuning data without human annotations using OCR and captioning tools. It discovered that resolution plays a significant role in recognizing textual information and explored several options. TGDoc (Wang et al., 2023b) improves LLaVAR and explores text-grounding for multimodal LLMs. Monkey (Li et al., 2023d) performed a surgery between simple text labels and high input resolution, enabling remarkable performance in visually-rich document images with dense text. TextMonkey (Liu et al., 2024c) has implemented shifted window attention to filter out similar tokens effectively. Meanwhile, DocPedia (Feng et al., 2023) and HRVDA (Liu et al., 2024a) have focused on enlarging input resolution to reduce the disparity between multimodal LLMs and visual document understanding. Recent works consider figures from academic papers as input, which are composed of text and figures (Li et al., 2024; Ye et al., 2023b). InternLM-XComposer2 (Dong et al., 2024) scales up the visual encoder's resolution to 4,096. OCR-based methods have been criticized for inducing more errors (Kim et al., 2022), which can now be alleviated with the help of large language models and visual encoders. LLaVA-Read uses PaddleOCR as a visual-text encoder because of its good generalization ability, and it can also use other visual encoders with great generalization ability.

**Visual Text Understanding**   Humans are incredibly robust to a variety of text permutations (Rayner et al., 2006) because they can leverage the graphical information in text (Sun et al., 2021). Previous work on visual language modeling aims to handle unseen out-of-vocabulary (OOV) words to overcome the drawback of a fixed vocabulary, which may lead to performance degradation (Kaddour et al.,

Figure 1: Model overview of LLaVA-Read, a multimodal LLM with dual encoders to handle both visual objects and texts. Given a text-rich image, the visual-text encoder extracts texts and their location information, feeding them to the OCR tokenizer. ViT-based low-resolution encoder (*e.g.*, 336×336) focuses on the global visual information and convolution-based encoder (*e.g.*, 768×768) focuses on visual details. The high-resolution encoder merges its information into low-resolution encoders, as not all details are useful in answering a question.

2023). PIXEL (Rust et al., 2022) achieved comparable performance with BERT (Devlin et al., 2018), but it can only perform natural language understanding tasks. Pixar (Tai et al., 2024) proposed the first pixel-based autoregressive LLM that performs text generation. (Gao et al., 2024) developed powerful screenshot LMs to unlock complex tasks such as chart understanding and UI navigation. Multimodal LLMs for text-rich images can extract visual texts, which is similar to the visual text understanding problem. The major difference is that multimodal LLMs not only need to comprehend visual texts but also visual objects and their relationship. Inspired by previous work (Gao et al., 2024), LLaVA-Read performs an visual text understanding analysis of multimodal LLMs on synthetic data, revealing their impressive capability on shorter scene text understanding but limited proficiency in comprehending large amounts of textual content within a text-rich image. This observation motivates us to add an additional visual-text encoder to enhance reading ability of multimodal LLMs.

## 3 LLaVA-Read: Enabling LLaVA to Read

LLaVA-Read is designed to enhance the comprehension of textual information within images, particularly in text-rich images. An overview of the model is shown in Figure 1. LLaVA-Read comprises multiple visual encoders, a visual-text encoder, and a large language model (LLM) serving as the decoder. Given an input image $\mathbf{X}_v$, the visual encoders generate visual features $\mathbf{Z}_v = f_v(\mathbf{X}_v)$, where $f_v$ consists of two visual encoders. Subsequently, we employ a multi-layer perceptron (MLP) projection $g$ to transform $\mathbf{Z}_v$ into visual tokens $\mathbf{H}_v = g(\mathbf{Z}_v)$ for the large language model. Notably, $\mathbf{H}_v$ shares the same embedding dimensions as the text tokens used by the LLM tokenizer. Different from the conventional architecture of multimodal large language models (Liu et al., 2023c), LLaVA-Read incorporates a visual-text encoder $f_t$ to better capture textual and layout information, along with a high-resolution encoder for finer visual details. The objective of the visual-text encoder is to extract text from an image, yielding visual-text tokens $\mathbf{H}_t = f_t(\mathbf{X}_v)$. Subsequently, we concatenate $\mathbf{H}_v$, $\mathbf{H}_t$, and $\mathbf{H}_q$, feeding them into the large language model to generate the desired response $\mathbf{Y}$.

In designing LLaVA-Read, we have the conviction that a visual encoder should specialize in processing visual objects, while a lightweight visual-text encoder should focus on extracting text within images. This approach, we believe, enhances the efficiency of the visual components, as text recognition presents distinct patterns compared to visual object detection. Although high-resolution visual encoders can capture finer details, they also generate a larger number of visual tokens. To mitigate additional computational costs associated with employing two visual encoders in LLaVA-Read, we merge the output of these encoders while maintaining the same visual tokens as in LLaVA. More details on architectural design are elaborated in Section 3.1. In essence, LLaVA-Read offers a multimodal LLM framework that leverages multiple visual encoders to improve visual token learning and conversion efficiency. To enhance collaborations between dual visual encoders, we propose layout-aware training during the two-stage training, as discussed in Sections 3.2 and 3.3.

## 3.1 MODEL ARCHITECTURE

**Visual-Text Encoder** Successful commercial visual-text extractor solutions typically have much smaller sizes compared to visual object detection models (Kirillov et al., 2023; Zou et al., 2024; Liu et al., 2023d). Increasing the resolution of the visual encoder for visual text recognition often incurs unnecessary computational costs, resulting in training and inference inefficiencies. While visual encoders excel at comprehending visual object information and scene texts, they often struggle with processing large chunks or paragraphs of visual text (further details in Section 4.1). Solutions such as Donut (Kim et al., 2022) and LayoutLM (Xu et al., 2020) offer neat approaches, but their generalization abilities are limited due to constraints in the pretraining dataset domains. Therefore, we consider employing open-source OCR tools as an alternative encoder to extract text and layout information. LLaVAR (Zhang et al., 2023d) initially utilize PaddleOCR [1] to construct a noisy pretraining dataset to enhance text recognition capabilities. Consequently, we integrate the lightweight PaddleOCR as our visual-text encoder. One major concern with the use of OCR-based methods is the potential for induced errors. However, collaboration between the visual encoder and the large language model mitigates this drawback. We use PaddleOCR as an option to verify our conviction on visual-text encoders. In addition, it demonstrates high efficiency in converting visual texts into text tokens for LLMs with great generalizability.

We use a customized OCR tokenizer to effectively encode both words and their respective locations (i.e., text bounding boxes). This tokenizer comprises a layout recovery module $f_r(\cdot)$ and a standard LLM tokenizer $f_q(\cdot)$. Upon receiving OCR results from a text-rich image, the layout recovery module $f_r$ processes the input by inserting spaces and line breaks, as described in (Wang et al., 2023a). The layout recovery process follows a heuristic approach: (*i*) Text boxes in the same row with detected words are identified and rearranged in top-to-bottom and left-to-right order based on their coordinates. (*ii*) The average character width is calculated for each row based on its width and word count. Placeholders are then inserted based on the horizontal distance between two text boxes in the same row, resulting in the extraction of single-row texts. (*iii*) Newline characters are inserted for each row, reconstructing the page layout. To avoid too many spaces being inserted, we have limited the number of spaces to a range of at least one and no more than ten, cutting the original number by half. Figure 8 in the appendix provides an example of how the OCR tokenizer operates. Once the plain text with layout information is obtained, it serves as part of the LLM prompts in both training and inference: $\mathbf{H}_t = f_t(\mathbf{X}_v) = f_q(f_r(f_{\text{OCR}}(\mathbf{X}_v)))$.

**Visual Encoders** LLaVA with low-resolution visual encoders has demonstrated significant success (Liu et al., 2023c), and the integration of a higher resolution encoder typically leads to performance improvements (Luo et al., 2024). However, high-resolution encoders tend to generate a larger number of visual tokens, and methods such as similarity-based token merging or compression may sacrifice details. Ideally, a high-resolution encoder should focus on question-related details without significantly increasing the number of visual tokens for language models. To address this, we propose a novel approach to merge details from high-resolution encoders to low-resolution encoders. Specifically, we utilize the pretrained OpenCLIP model `ConvNext-L/32-320` as the high-resolution encoder $f_h$ and the pretrained CLIP model `ViT-L/14-336` as the low-resolution encoder $f_s$. The high-resolution visual encoder with an image patch size of 32 can accommodate approximately 2.3 times higher resolution images compared to the low-resolution encoder with a patch size of 14. For example, if the low-resolution encoder takes the image $\mathbf{X}_{v_s}$ with dimensions $336 \times 336$, then the high-resolution encoder processes the image $\mathbf{X}_{v_h}$ with dimensions $768 \times 768$. Position embedding interpolation will be applied for encoders if the resolution is higher than 768.

To better capture visual details, we perform layer-wise fusion, which involves embedding high-resolution features into the low-resolution visual pathway through mapping modules and dynamic scoring (Luo et al., 2024). The fusion process occurs at different layers of the Vision Transformer (ViT), ensuring that the low-resolution features also contain rich semantics. In more detail, we merge the high-resolution visual encoder features $f_h(\mathbf{X}_{v_h})$ into the low-resolution encoder $f_s(\mathbf{X}_{v_s})$. Both visual encoders have 12 layers and we perform the feature merging for 1st, 4th, and 7th layers. For a specific layer $l$, we calculate the patch-wise weight score $g_l = f_g([f_{s_l}(\mathbf{X}_{v_s}), f_{h_l}(\mathbf{X}_{v_h})])$, where $f_g$ is a projection layer. Then we merge them $f_{s_l}(\mathbf{X}_{v_s}) + g_l \cdot f_m(f_{h_l}(\mathbf{X}_{v_s}))$, where $f_m$ is a projection layer with the same input and output dimensions.

---

[1] https://github.com/PaddlePaddle/PaddleOCR/blob/main/README_en.md

To prevent the generation of additional visual tokens, we combine the visual features of both visual encoders as follows: $f_v(\mathbf{X}_v) = f_h(\mathbf{X}_{v_h}) + f_s(\mathbf{X}_{v_s})$, where we first perform a linear projection on the final-layer features with the same input and output dimensions and then add these two features together. It ensures that the resulting visual tokens of LLaVA-Read are of the same number as standard LLaVA, avoiding lossy token compressions (Liu et al., 2024c). This straightforward merging strategy proves to be effective in text-rich image understanding, as demonstrated in Section 4.

### 3.2 LAYOUT-AWARE PRETRAINING FOR FEATURE ALIGNMENT

Starting with the LAION-5B dataset, we selectively retain images prominently featuring text. From the filtered LAION-5B, a random sample of 10,000 images is clustered into 50 groups based on CLIP-ViT-B/32 visual features (Zhang et al., 2024). After careful examination of the clustering results, 14 clusters are meticulously chosen, encompassing diverse text-rich images such as posters, book covers, advertisements, and educational documents. In the pretraining stage, we utilize the LLaVA LCS-558k pretraining dataset, mainly comprising natural images. Furthermore, we augment this dataset by incorporating 422k LAION images from LLaVAR (Zhang et al., 2023d), 99k slides images from TGDoc (Wang et al., 2023b), and 112k document-related images from various public datasets. Table 6 in the appendix shows detailed statistics of the training data. Similar to LLaVA , only the projection layer is trained during the pretraining stage. The visual-text encoder is not utilized in the pretraining stage unless explicitly mentioned.

**Task I: Text Recognition**   Following LLaVAR (Zhang et al., 2023d), we use PaddleOCR to extract visual texts from the original images and concatenated all detected words to form the target sequence. We then generated single-turn conversations for each image by (i) randomly sampling an input instruction and (ii) using the recognized text sequence as the desired output response. It is worth noting that such instruction-following data may be noisy due to the varying performance of OCR tools across different fonts and backgrounds.

**Task II: Text Localization**   The text recognition task extracts text information only but ignores PaddleOCR layout information. Similar to Task I, we created single-turn conversations for each image by (i) randomly sampling an instruction to extract both texts and bounding boxes and (ii) using the recognized text sequence along with its bounding boxes as the desired output response. This simple training scheme is effective and allows the model to develop grounding ability (You et al., 2023). It is important to represent bounding boxes accurately; therefore, we converted each integer value of box coordinates into a float value, ranging from 0 to 1. In addition, we used the top-left and bottom-right coordinates to represent the text boxes.

**Task III: Page Parsing**   To better capture layout information, we pretrain the model to parse image pages into plain text with minimal loss of layout information. We adopt the layout reconstruction module $f_r(\cdot)$ to parse both words and bounding boxes, incorporating placeholders and new-line characters to reconstruct the image layout (Wang et al., 2023a). For example, we utilize images from PlotQA (Methani et al., 2020) and ChartQA (Masry et al., 2022), using the source data to construct the corresponding Markdown codes. More details are provided in Appendix B.1.

**Task IV: Layout Recovery**   The layout reconstruction task aims to transfer the ability of $f_r(\cdot)$ to LLaVA-Read. It utilize OCR results from Task II and parse pages as in Task III to build instruction tuning pairs. This task is designed to teach the language model to better comprehend coordinates and reconstruct the layout using visual-text results. Representative examples of different pretraining tasks are provided in Figure 7 and Figure 8 in the Appendix B.1.

### 3.3 LAYOUT-AWARE FINETUNING FOR INSTRUCTION FOLLOWING

Jointly understanding both visual texts and objects is crucial to efficiently analyzing text-rich images. To enhance the model's visual object understanding, we perform finetuning using the natural image finetuning dataset from LLaVA. Although scaling up the dataset could potentially further improve visual object understanding, we did not explore this direction in this paper. To improve the understanding of visual texts and align different encoders, we combine instruction tuning datasets from LLaVAR (Zhang et al., 2023d), TGDoc (Wang et al., 2023b), and TRINS (Zhang et al., 2024) for text-rich image instruction tuning. Additionally, we merge visual question-answering data sets related to documents from various sources (Mathew et al., 2022; 2020; Pasupat & Liang, 2015; Masry et al., 2022) to enhance performance. In total, we assemble around 425k instruction finetuning datasets.

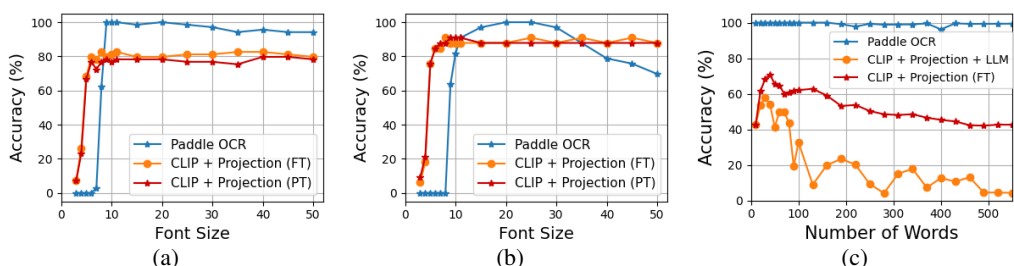

Figure 2: Comparison of word recognition accuracy among different methods using (a) multiple font dimensions against a plain background (b) multiple font dimensions against a natural image background (c) varying word counts.

In the finetuning stage, both the low- and high-resolution visual encoders are kept frozen. We continue to finetune projection layers and the base large language model to better align tokens from visual encoders and the visual-text encoder. We consider two scenarios in the finetuning stage: *i*) For natural image training, only visual tokens $\mathbf{H}_v$ and question tokens $\mathbf{H}_q$ are used, following the LLaVA training scheme (Liu et al., 2023c). *ii*) For text-rich image training, the visual-text encoder is additionally used to extract words and boxes, facilitating the recovery of their layouts: $\mathbf{H}_t = f_q(f_r(f_{\text{OCR}}(\mathbf{X}_v)))$. The training target is the expected response $\mathbf{Y}$ from the instruction tuning set.

## 4 EXPERIMENTAL RESULTS

We first perform a visual text understanding analysis, which inspires us to propose the visual-text encoder branch in LLaVA-Read. Then, we evaluate the performance of LLaVA-Read on classical text-rich image benchmarks and OCRBench Liu et al. (2023e). We pretrain our model for 1 epoch to obtain projection layers with a batch size of 128, a context window size of 2048, and a learning rate of $2e$-3. We further fine-tune LLaVA-Read on the 425k instruction tuning set for 1 epoch with a learning rate of $2e$-5 with a batch size of 32 and a context window size of 4096. We use Vicuna-1.5 13B as the base language model. All experiments were performed on NVIDIA A100s.

### 4.1 VISUAL TEXT UNDERSTANDING ANALYSIS

**Settings**  Following previous work (Rust et al., 2022; Tai et al., 2024; Gao et al., 2024), we generate synthetic data to evaluate the text recognition ability of different visual encoders by varying font sizes and number of words, as shown in Figure 2. We use PaddleOCR as a simple and effective visual-text encoder and OpenAI CLIP plus trained projection layers to inspect the text recognition ability of visual encoders. We use multiple fonts to render text-rich images and use the OCR accuracy as a metric. For PaddleOCR and multimodal LLM, accuracy means that the rendered ground-truth words can be exactly found in the outputs. For CLIP with projection, we first obtain the model outputs, which are visual token embeddings, and then perform similarity-based ranking with words from the language model's vocabulary. If the ground truth words can be found in the top-3 words, we count these are detected by the model. We list a few research questions to help the reader better understand our experimental results. Please note that we removed stop-words from the NLTK (Bird et al., 2009) package as many repeat stop-words exist in text paragraphs.

**RQ1: How many pixels do we need to recognize words?**  We first investigate the performance of different modules on text recognition ability with different font sizes. In Figure 2a, all text-rich rendered images have a plain white background, which is similar to the scan document images or screen shots. In Figure 2b. All rendered text-rich images are rendered with a random selected image as the background, corresponding to the scene text and poster settings. In both scenarios, we use the terms of machine learning as the texts to recognize, each phrase containing no more than four words. We measure the font size with its vertical heights. CLIP with projection can recognize texts with a minimum font size at 6 pixels to achieve its best performance. In addition, the CLIP with projection performance is similar before and after the fine-tuning stage.

> **Finding 1.**  Multimodal LLMs equipped with traditional visual encoders excel at understanding shorter scene text but struggle with dense textual content in text-rich images.

Table 1: Model performance (accuracy %) on text-based VQA. We use † to refer to the results obtained from previous work Liu et al. (2023e).

| | ST-VQA | TextVQA | DocVQA | ChartQA | InfoVQA | FUNSD | SROIE |
|---|---|---|---|---|---|---|---|
| BLIP-2 (Li et al., 2023c) † | 21.7 | 32.2 | 4.9 | 3.4 | 11.3 | 0.20 | 0.14 |
| MiniGPT4 (Zhu et al., 2023) † | 14.0 | 18.7 | 3.0 | 4.3 | 13.3 | 1.19 | 0.04 |
| LLaVA-1.5 (Liu et al., 2023c) † | 38.1 | 38.7 | 8.5 | 9.3 | 14.7 | 0.20 | 1.70 |
| LLaVAR (Zhang et al., 2023d) † | 39.2 | 48.5 | 11.6 | 12.2 | 16.5 | 0.50 | 5.20 |
| mPLUG-Owl2 (Ye et al., 2023c) † | 29.3 | 40.3 | 6.9 | 19.4 | 18.9 | 1.40 | 3.20 |
| Monkey (Li et al., 2023d)† | 54.7 | 64.4 | 50.1 | 54.0 | 25.8 | 24.1 | 41.9 |
| LLaVA-1.6-8B (Liu et al., 2024b) | 48.3 | 54.7 | 41.2 | 43.0 | 21.2 | 24.8 | 36.9 |
| LLaVA-1.6-34B (Liu et al., 2024b) | 50.3 | 58.3 | 52.5 | 58.6 | 32.6 | 28.9 | 42.5 |
| Idefics-2 (Laurençon et al., 2024) | 56.8 | **66.6** | 60.4 | 76.3 | 29.9 | 31.1 | 52.8 |
| Cambrian-1 (Tong et al., 2024) | 59.3 | 64.0 | 56.7 | 77.8 | 29.4 | 28.7 | 39.8 |
| Eagle (Shi et al., 2024) | 57.8 | 66.2 | 68.5 | **82.1** | 40.1 | **39.7** | 48.8 |
| LLaVAR w/ OCR | 49.2 | 54.9 | 48.3 | 25.6 | 28.4 | 23.2 | 36.6 |
| LLaVA-Read w/o OCR | 54.0 | 60.1 | 47.8 | 65.0 | 29.4 | 22.0 | 35.0 |
| LLaVA-Read w/o Visual | 23.5 | 32.4 | 43.9 | 27.6 | 24.0 | 16.5 | 45.1 |
| LLaVA-Read | **59.5** | 66.0 | **71.3** | 76.3 | **40.2** | 36.7 | **61.9** |

**RQ2: Is one text token worth one visual token?**   In Figure 2c, we show the performance of three different modules on text recognition ability. When the number of words is less than 50, the visual encoder with projection and Multimodal LLM (*i.e.*, CLIP + Projection + LLM) can work, but with lower accuracy. However, when there are large chunks of texts, *i.e.*, the number of words becomes larger, the performance of both modules starts to collapse. We observe similar trends in various multimodal LLMs as shown in Appendix A. This analysis shows the low efficiency of using the CLIP encoder to transform visual texts into visual tokens, and language models can only handle short sequences of visual tokens with textual information. In contrast, the visual text encoders (*i.e.*, PaddleOCR) shows much better and consistent performance in encoding large chunks of visual texts, underscoring its essential role of multimodal LLMs for great reading capabilities.

> **Finding 2.**   Traditional visual encoders generate fixed-length visual tokens, leading to inefficient token use when converting visual texts into visual tokens for language models.

**RQ3: Is the visual-text encoder always the best in text recognition?**   The visual-text encoder we used in experiments is PaddleOCR, a model that is considerably more compact (less than 1%) compared to OpenAI CLIP `ViT-L/14-336`. PaddleOCR is great at recognizing large chunks of text, but it requires a minimum of 9 pixels and cannot recognize texts smaller than 7 pixels (in terms of the height of characters), while CLIP + Projection can do better. In the scene text experiment (Figure 2b), font size does not affect the performance of CLIP with projection when the font size increases, while PaddleOCR gets worse. In summary, a visual text encoder such as PaddleOCR proves to be beneficial, and a visual encoder can also help to understand visual text in certain cases.

> **Finding 3.**   PaddleOCR serves as a simple visual text encoder with **adaptive** context lengths, offering great token efficiency. Although its smaller size may lead to errors, these can be mitigated by large language models.

## 4.2 MAIN RESULTS

We evaluate the LLaVA-Read and its baselines on OCRBench and other text-rich image benchmarks [2] in Table 1 and Table 3(a). LLaVA-Read shows state-of-the-art performance in the OCR bench among open-source models and comparable performance with Gemini and GPT-4v. Compared with other baselines, LLAVA-Read with low-resolution encoders can beat Text-Monkey, the best open-source model with a large gap, showing the benefits of adding visual-text encoders. Specifically, performance on KIE and other classical document VQA is greatly improved, where large chunks of text are embedded in images. In addition, LLaVA-Read with combined higher resolution encoder (*i.e.*, LLaVA-Read-H) further improves the performance of the model, especially on ChartVQA and TextVQA. For ChartVQA, adding layout information improves 30% performance improvement

---

[2]Evaluation protocol from OCRBench Liu et al. (2023e), which uses different metrics for QA accuracy.

Table 2: LLaVA-Read can extract information from the image and answer following the required format, despite a few errors compared with GPT-4V and LLaVA-1.5.

| Visual input example, Constrained JSON Output: | |
|---|---|
| 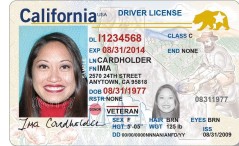 | **User**: Please read the text in this image and return the information in the following JSON format (note xxx is placeholder, if the information is not available in the image, put "N/A" instead). {"class": xxx, "DLN": xxx, "DOB": xxx, "Name": xxx, "Address": xxx, "EXP": xxx, "ISS": xxx, "SEX": xxx, "HGT": xxx, "WGT": xxx, "EYES": xxx, "HAIR": xxx, "DONOR": xxx} |
| GPT-4V | {"class": "C", "DLN": "1234568", "DOB": "08/31/1977", "Name": "Ima Cardholder", "Address": "2570 24th Street, Anytown, CA 95818", "EXP": "08/31/2014", "ISS": "08/31/2011", "SEX": "F", "HGT": "5-05", "WGT": "125 lbs", "EYES": "BRN", "HAIR": "BRN", "DONOR": "VETERAN"} |
| LLAVA 1.5 | {"class": "C", "DLN": "1132456789", "DOB": "08/23/1971", "Name": "Ima Cardholder", "Address": "Anytown, State, Zip", "EXP": "08/23/2014", "ISS": "California", "SEX": "F", "HGT": "5'5", "WGT": "123", "EYES": "Brown", "HAIR": "Brown", "DONOR": "N/A"} |
| LLaVA-Read | {"class": "C", "DLN": "1234568", "DOB": "08/31/1977", "Name": "Ima Cardholder", "Address": "2570 24th Street, Anytown, CA 95818", "EXP": "08/31/2014", "ISS": "08/31/2009", "SEX": "F", "HGT": "5-05", "WGT": "125", "EYES": "Brown", "HAIR": "Brown", "DONOR": "VETERAN"} |

Table 3: (a) Results of Multimodal LLMs on OCRBench. (b) Ablation results of multimodal LLMs in OCRBench. Recog. represents text recognition, $VQA^S$ represents Scene Text-Centric VQA, $VQA^D$ represents Document-Oriented VQA.

| Method | Recog. | $VQA^S$ | $VQA^D$ | KIE | Total |
|---|---|---|---|---|---|
| Gemini | **215** | **174** | 128 | 134 | **651** |
| GPT-4v | 167 | 163 | **146** | **160** | 636 |
| idefics-2 | **248** | **174** | 106 | 110 | 638 |
| MiniCPM-v2 | 245 | 171 | 103 | 86 | 605 |
| DocOwl 1.5 | 182 | 157 | 126 | 134 | 599 |
| Eagle | 152 | 151 | **131** | 134 | 569 |
| Text-Monkey | 169 | 164 | 115 | 116 | 561 |
| LLaVA1.6-8B | 218 | 147 | 100 | 114 | 579 |
| mPLUG-Owl2 | 153 | 153 | 41 | 19 | 366 |
| LLaVA1.5-13B | 176 | 129 | 19 | 7 | 331 |
| LLaVA-Read | 234 | 167 | 125 | **145** | **671** |

(a)

| Method | Res. | $VQA^S$ | $VQA^D$ | KIE |
|---|---|---|---|---|
| LLaVA + OCR | 336 | 147 | 85 | 105 |
| LLaVA-Read | 336 | 151 | 101 | 145 |
| w/o Layout FT | 336 | 150 | 90 | 116 |
| LLaVA-Read | 768 | **170** | 108 | 145 |
| LLaVA-Read | 1024 | 167 | **125** | **145** |
| w/o OCR | 1024 | 151 | 100 | 97 |
| w/o Visual | 1024 | 78 | 56 | 116 |
| w/o task II. | 1024 | 160 | 110 | 140 |
| w/o task III | 1024 | 162 | 106 | 142 |
| w/o task IV. | 1024 | 158 | 106 | 146 |
| w/o Doc. FT | 1024 | 165 | 99 | 143 |

(b)

in terms of QA accuracy. When adding high-resolution visual encoders, the model performance improves further by about 20%. The layout information within a chart image is too complex to reconstruct with a heuristic function, and a high-resolution visual encoder can help in this case. For TextVQA, it shows the importance of visual encoders in scene text understanding as the performance becomes better as the resolution of visual encoders increases. This observation is consistent with what we find in Section 4.1. We also evaluate two variants of LLaVA-Read, which only uses visual encoders or the visual text encoder. It is evident that there are two distinct types of text-rich images. For images with more visual elements and complex layouts, such as scene text and charts, visual encoders are crucial. In contrast, for text-rich images with dense text and plain backgrounds, such as tables, forms, and scanned documents, visual text encoders play a more significant role. Hence, it is difficult to use a single encoder to handle all text-rich images (Tong et al., 2024; Shi et al., 2024).

**Generated Examples** Figure 3 shows a generated example of LLaVA-Read on complex infographics. Table 2 shows another example, for which LLaVA-Read needs first parse this image and then output results in the JSON format following the scheme in the user instruction. LLaVA-Read correctly extracts all the information from the given image, while LLaVA 1.5 and GPT-4V still make minor mistakes. More generated examples of grounding are provided in the Appendix C.

**Ablation Study on Text-rich Image VQA** We first compare LLaVA-Read with LLaVA plus OCR, where OCR words are provided to LLaVA in the training. The gap between these two settings shows the benefits of the OCR tokenizer, where both OCR texts and boxes are used. LLaVA-Read w/o layout finetuning still shows better performance compared with LLaVA + OCR, validating the effectiveness of layout-aware pretraining. We also perform another ablation study on layout pretraining; LLaVA-Read models with specific pretraining tasks removed all show inferior performance. If we remove the 100k document-related finetuning dataset, the performance on document-oriented VQA will decrease. We find that the model usually fails on the ChartVQA after we manually inspect the results. The

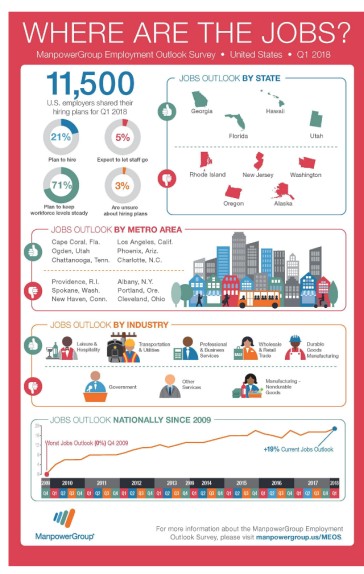

Which metro area in California that has a good job outlook?

The metro area in California with a good job outlook is Los Angeles.

What are the industries that have a good job Outlook?

The industries with a good job outlook according to the image are Leisure, Hospitality, Transportation, Utilities, Professional, Business Services, Wholesale, Retail Trade, and Durable Goods Manufacturing.

What are key take aways from this diagram?

The key take aways from the diagram are that the job outlook is positive since 2009, with an increase in jobs reported in several industries and regions. The diagram also highlights that hiring plans are up across the nation, and there is a breakdown of job outlook by state, metro area, and industry.

What is 21% for?

The 21% figure represents the percentage of U.S. employers who plan to hire additional staff in Q1 2018.

Figure 3: An example that showcases complex reasoning in infographics. It shows LLaVA-Read can comprehend both visual texts and objects within a sophisticated layout.

resolution of the visual encoder plays an important role in multimodal LLM since higher resolution usually means more details. If we add high-resolution visual encoder, we observe improvement on both scene text-centric VQA and document-oriented VQA. Furthermore, if we increase the resolution from 768 to 1024, the performance is enhanced. Removing the PaddleOCR from LLaVA-Read does not cause a model collapse but leads to performance degradation.

**Ablation study on Text Recognition**    Table 4 shows the results of different methods in OCRBench text recognition tasks. The text recognition task includes six subsets: *i*) Regular Text Recognition, *ii*) Irregular Text Recognition, *iii*) Artistic Text Recognition, *iv*) Handwriting Recognition, *v*) Digit String Recognition, and *vi*) Non-Semantic Text Recognition. Each subset has 50 test examples, and the total number of test examples is 300. PaddleOCR is the worst one and only works well on regular text recognition and non-semantic random text recognition. Spelling errors or missing characters are the main reason for the poor performance of PaddleOCR. For three LLaVA-Read variants, models with higher resolution usually have better performance. If we remove the support of PaddleOCR, LLaVA-Read still works with slightly worse performance. However, as indicated in Table 4, the performance of LLaVA-Read in VQA significantly declines when OCR support is removed.

Table 4: Ablation Results on Text Recognition from OCR Bench.

| Method | Res. | Reg. | Irreg. | Hand. | Art. | Digit. | Non-Sem. | Total |
|---|---|---|---|---|---|---|---|---|
| PaddleOCR | 960 | 40 | 20 | 21 | 2 | 8 | 49 | 140 |
| LLaVA-Read | 336 | 48 | 43 | 43 | 34 | 14 | 24 | 206 |
| LLaVA-Read | 1024 | 48 | 42 | 40 | 18 | 25 | 47 | 220 |
| w/o OCR | 1024 | 46 | 41 | 41 | 18 | 20 | 28 | 194 |

**Finding 4.**    Multimodal LLMs can recognize visual words, but they do not exhibit the same level of understanding when these words appear in text inputs.

## 5 CONCLUSIONS

In this paper, we first analyze the visual text understanding ability of multimodal large language models, demonstrating the essential need for integrating extra visual text encoders. Then we propose LLaVA-Read, a model architecture that enhances the reading ability of multimodal large language models (LLMs) by integrating layout information and using multiple visual encoders. Through a comprehensive evaluation on text-rich image understanding tasks, LLaVA-Read outperforms existing state-of-the-art models, demonstrating the effectiveness of incorporating layout information and utilizing multiple visual encoders in improving the comprehension of textual content situated in images. This work contributes to the advancement of multimodal language models and provides valuable insights for further research in enhancing the reading ability of such models.

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

# A   VISUAL TEXT UNDERSTANDING ANALYSIS DETAILS

| # of tokens | 10 | 50 | 100 | 220 | 340 | 550 |
|---|---|---|---|---|---|---|
| PaliGenMa-3B-896 | 71.4 | 34.5 | 20.7 | 12.7 | 9.0 | 9.5 |
| Phi-3v-mini-128k | 57.1 | 6.9 | 24.1 | 5.2 | 17.8 | 15.0 |
| LLaVA-Next-Mixtral-7B | 57.1 | 43.5 | 50.0 | 15.7 | 11.4 | 17.2 |
| LLaVA-Next-Vicuna-13B | 57.1 | 48.3 | 32.8 | 20.9 | 10.4 | 10.7 |
| LLaVA-Next-Yi-34B | 71.4 | 62.1 | 48.3 | 29.9 | 23.8 | 11.0 |

Table 5: Token accuracy of different Multimodal LLMs across different numbers of tokens

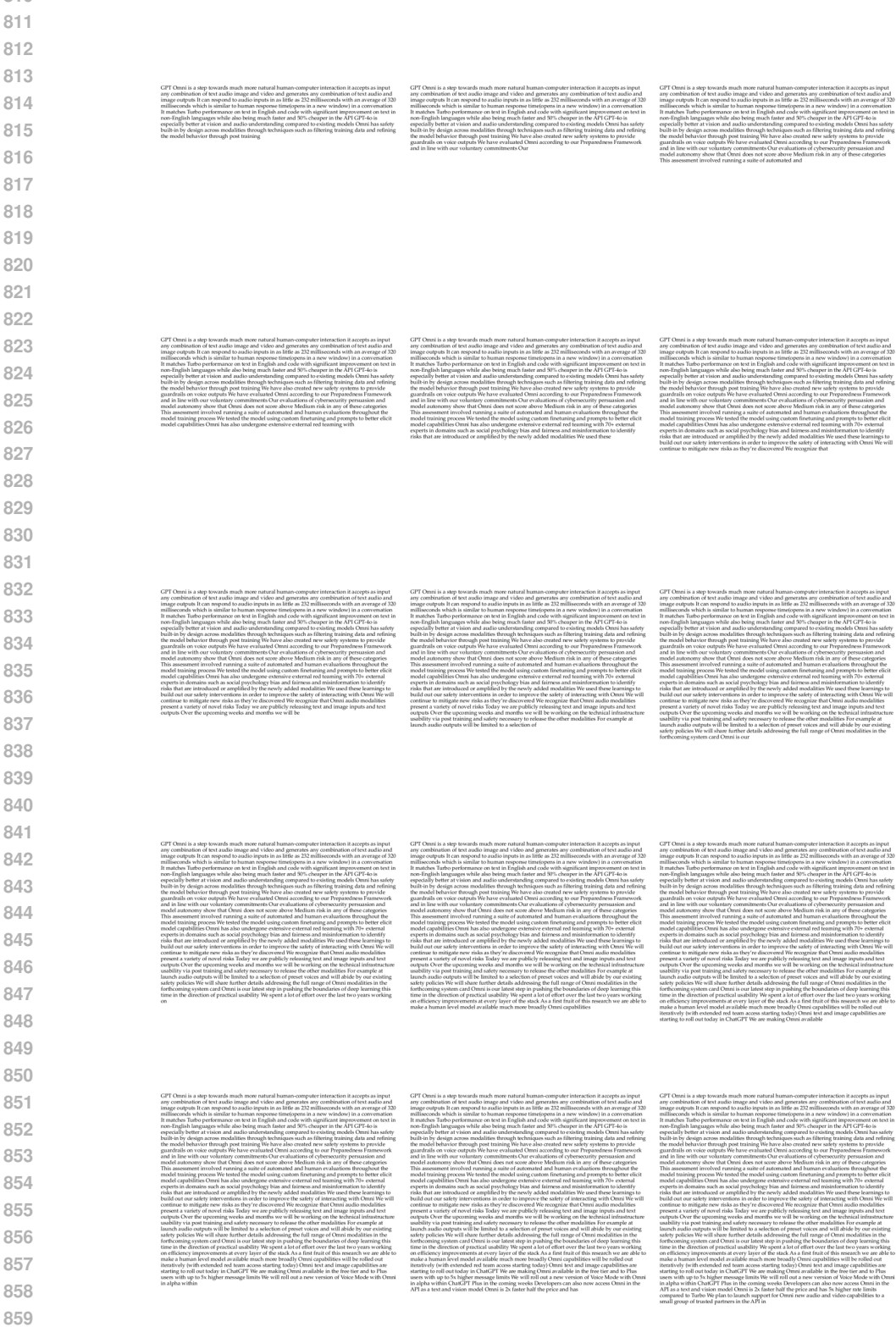

Figure 4: Different length of dense texts with plain background.

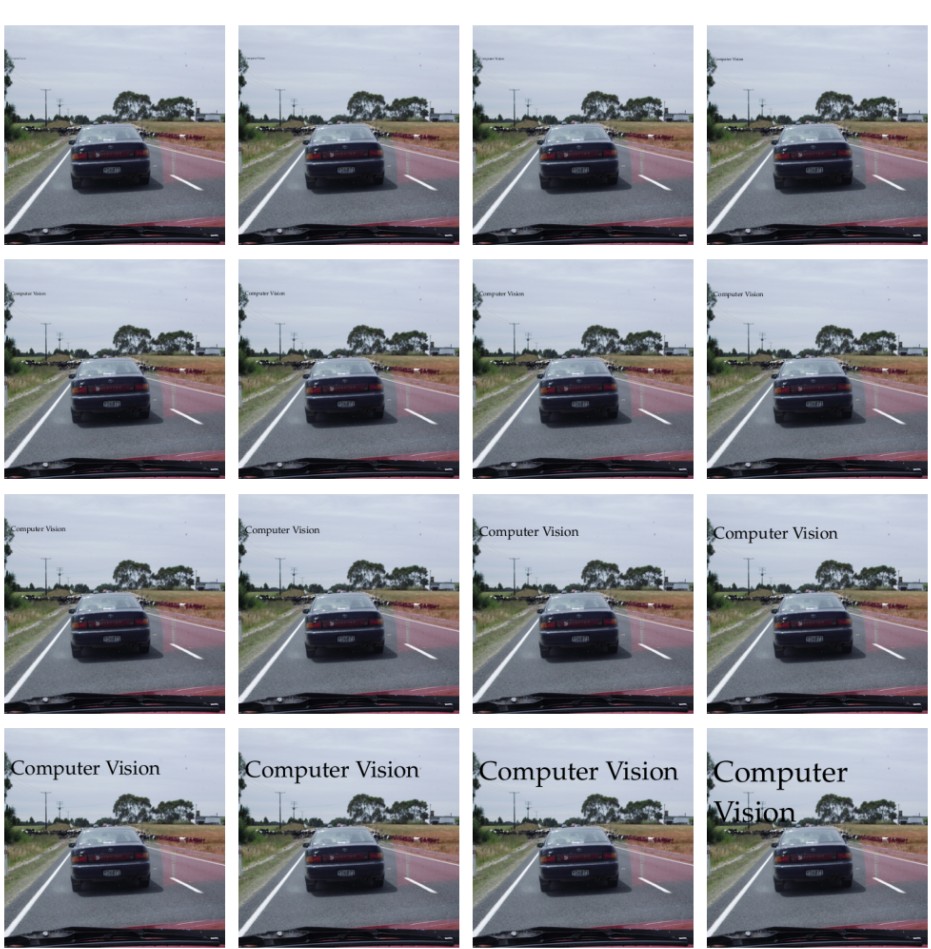

Figure 5: Different font sizes with natural image background.

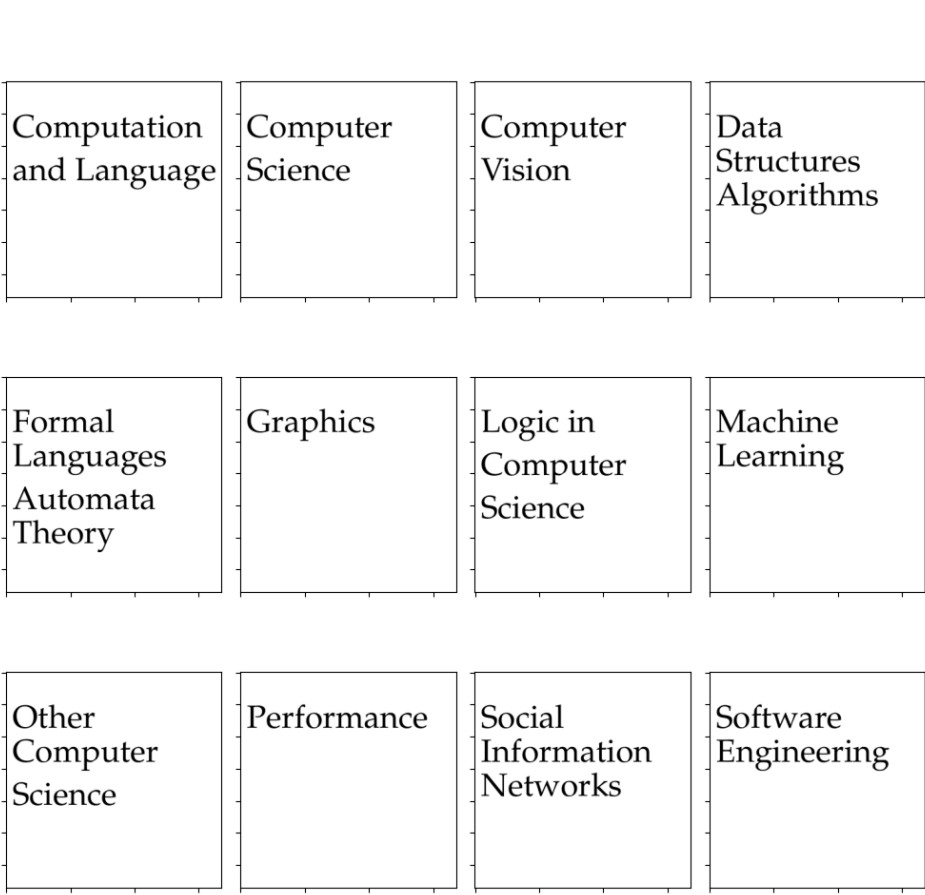

Figure 6: Different ML terms with plain background.

# B  TRAINING DATA DETAILS

Table 6: Dataset statistics for layout-aware pretraining and finetuning.

| Dataset | Sources | Size | Annotation Type |
|---------|---------|------|-----------------|
| LCS-558k | LLaVA-1.5 Liu et al. (2023a) | 558k | Caption (🤖) |
| Text Recognition | LLaVAR Zhang et al. (2023d) | 422k | OCR words (🤖) |
| Text Localization | TGDoc Wang et al. (2023b) | 465k | OCR words and boxes (🤖) |
| Layout Recovery | LLaVAR Zhang et al. (2023d) | 287k | OCR-based text layout (🤖) |
| Page Parsing | LLaVAR, Table & Chart | 509k | Text layout (👥 + 🤖) |
| LLaVA-FT | LLaVA-1.5 Liu et al. (2023a) | 150k | VQA (🤖) |
| LLaVAR-FT | LLaVAR Zhang et al. (2023d) | 16k | VQA (🤖) |
| TRINS-QA | TRINS Zhang et al. (2024) | 100k | VQA (👥 + 🤖) |
| TRINS-Cap | TRINS Zhang et al. (2024) | 35k | Caption (👥) |
| Text-Grounding | TGDoc Wang et al. (2023b) | 12k | VQA (🤖) |
| Doc-related VQA | Multiple Sources | 112k | VQA (👥) |

## B.1  PRETRAINING DATA EXAMPLES

We present pretraining instruction templates of Task II in Table 8, Task III in Table 7 and Task IV in Table 9. Pretraining examples randomly selected are shown in Figure 7 and 8.

## B.2  FINETUNING DATA EXAMPLES

The finetuning examples randomly selected are shown in Figure 7 and 8.

| No. | User Instruction |
|-----|------------------|
| 1 | Could you locate the text in the image and furnish the coordinates [xmin, ymin, xmax, ymax] for each text block? |
| 2 | Please recognize all the text within the image and supply the coordinates [xmin, ymin, xmax, ymax] for each text element. |
| 3 | Can you identify and extract all the text from the image, and include the coordinates [xmin, ymin, xmax, ymax] for each text block? |
| 4 | I would like you to recognize the text within the image and provide the bounding box [xmin, ymin, xmax, ymax] for each piece of text. |
| 5 | Kindly identify and extract text from the image, and supply the coordinates [xmin, ymin, xmax, ymax] for each text portion. |
| 6 | Can you recognize all the text present in the image and provide the corresponding bounding boxes or coordinates [xmin, ymin, xmax, ymax]? |
| 7 | I'm looking for you to detect and list all text within the image, accompanied by their bounding box coordinates [xmin, ymin, xmax, ymax]. |
| 8 | Please analyze the image for text, and for each text segment, provide the bounding box coordinates [xmin, ymin, xmax, ymax]. |
| 9 | I'd appreciate it if you could identify and provide the coordinates [xmin, ymin, xmax, ymax] for all text found in the image. |
| 10 | Kindly pinpoint the text in the image and provide the coordinates [xmin, ymin, xmax, ymax] for each text block. |

Table 7: Task II: Text Localization Templates

# C  MORE GENERATED EXAMPLES

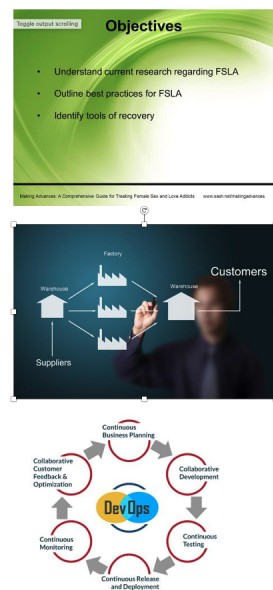

Can you identify and extract all the text from the image, and include the coordinates [xmin, ymin, xmax, ymax] for each text block?

Objectives [0.352, 0.042, 0.648, 0.128] Understand current research regarding FSLA [0.154, 0.289, 0.861, 0.344] Outline best practices for FSLA [0.158, 0.404, 0.643, 0.448] Identify tools of recovery [0.158, 0.516, 0.541, 0.568] Making Advances: A Comprehensive Guide for Treating Female Sex and Love Addicts [0.023, 0.932, 0.697, 0.966] www.sash.net/makingadvances [0.723, 0.935, 0.961, 0.961]

Can you recognize all the text present in the image and provide the corresponding bounding boxes or coordinates [xmin, ymin, xmax, ymax]?

Factory [0.345, 0.159, 0.418, 0.201] Customers [0.747, 0.250, 0.957, 0.310] Warehouse [0.113, 0.359, 0.217, 0.388] Warehouse [0.595, 0.349, 0.700, 0.378] Suppliers [0.094, 0.760, 0.231, 0.815]

Can you recognize all the text present in the image and provide the corresponding bounding boxes or coordinates [xmin, ymin, xmax, ymax]?

Continuous Business Planning [0.344, 0.117, 0.556, 0.193] Collaborative Customer Feedback& Optimization [0.077, 0.279, 0.236, 0.422] Collaborative Development [0.641, 0.310, 0.803, 0.388] DevOps [0.344, 0.458, 0.540, 0.560] Continuous Monitoring [0.101, 0.633, 0.243, 0.711] Continuous Testing [0.654, 0.628, 0.794, 0.698] Continuous Release and Deployment [0.341, 0.820, 0.573, 0.891]

Figure 7: Pretraining Examples for Task II, which is produced by PaddleOCR.

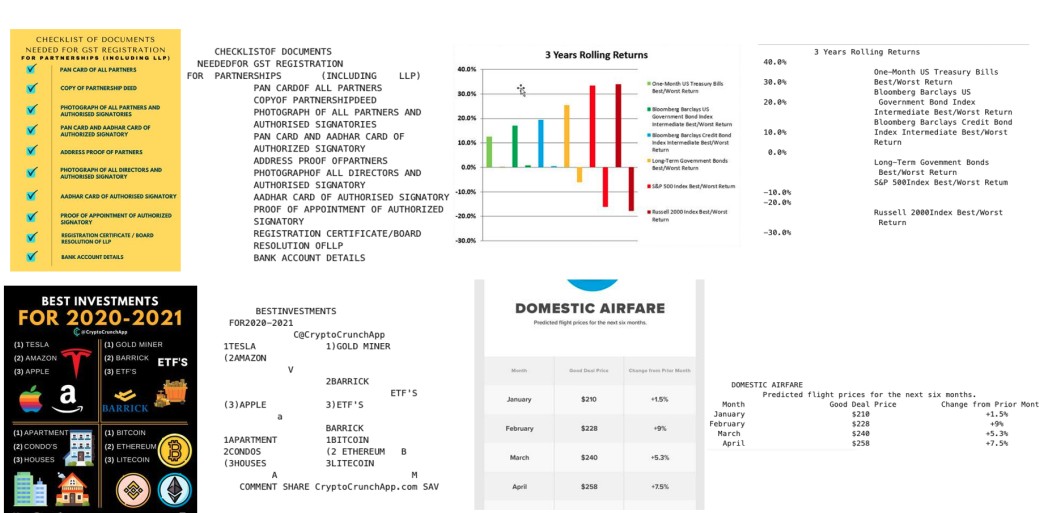

Figure 8: Pretraining Examples for Task III and IV, which is produced by PaddleOCR and OCR Tokenizer.

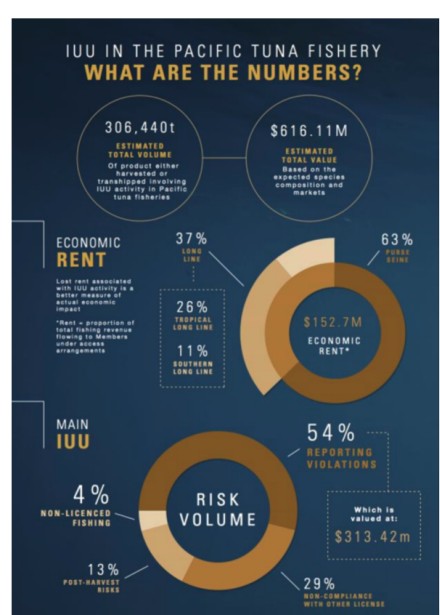

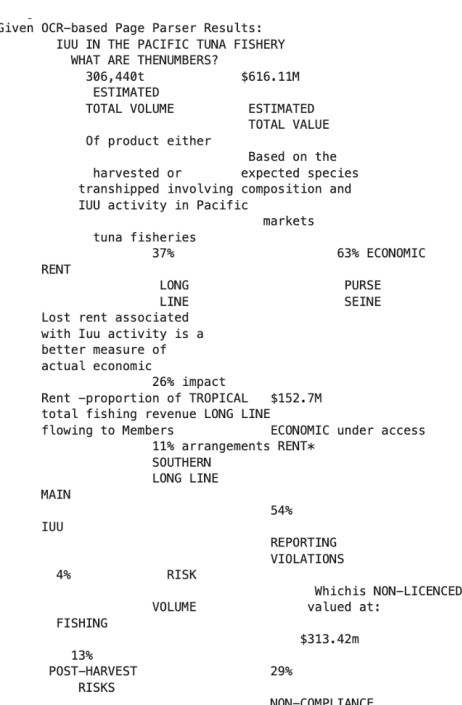

Q: Which are two types of economic rents?
A: Long Line, Purse Siene

Figure 9: A Layout-aware Finetuning Example of Infographics.

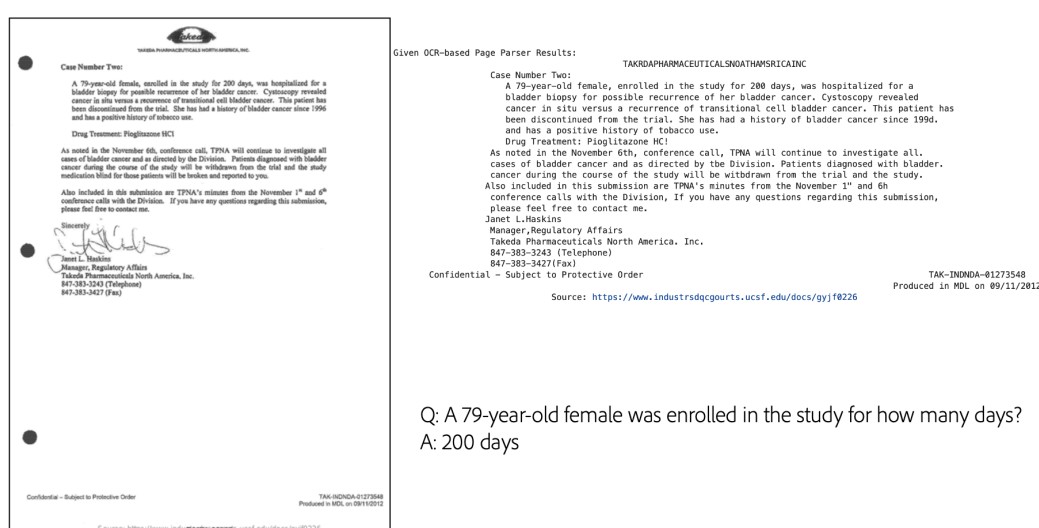

Q: A 79-year-old female was enrolled in the study for how many days?
A: 200 days

Figure 10: A Layout-aware Finetuning Example of Document images.

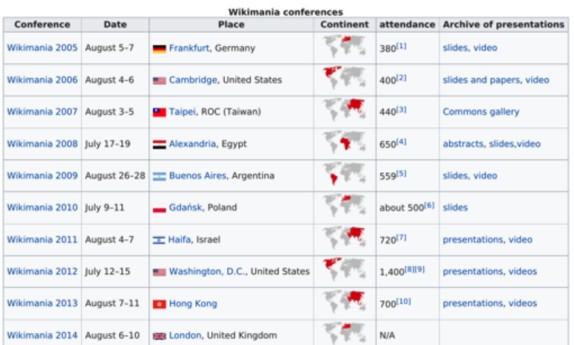

Q: What was the only conference to have an attendance over 1,000?
A: Wikimania 2012

```
Given OCR-based Page Parser Results:
                              Wikimania conferences
   Conference     Date          Place          Continent. attendance Archive of presentations
 Wikimania 2005 August 5-7  Frankfurt, Germany         380[1]    slides, video
 Wikimania 2006August 4-6   Cambridge, United States   400[2]    slides and papers, video
 Wikimania 2007 August 3-5  Taipei, ROc (Taiwan)       440[3]     Commons gallery.
 Wikimania 2008 July 17-19  Alexandria, Egypt          650[4]    abstracts, slides,video
 Wikimania 2009 August 26-28 . Buenos Aires, Argentina 559[5]    slides, video
 Wikimania 2010July 9-11    Gdansk, Poland          about 500[6] slides
 Wikimania 2011 August 4-7- Haifa, Israel              720[7]    presentations, video
 Wikimania 2012July 12-15  Washington, D.C., United States. 1,400[8][9] presentations, videos
 Wikimania 2013August 7-11   Hong Kong                 700[10]   presentations, videos
 Wikimania 2014 August 6-10 * London, United Kingdom   N/A
```

Figure 11: A Layout-aware Finetuning Examples of Table images

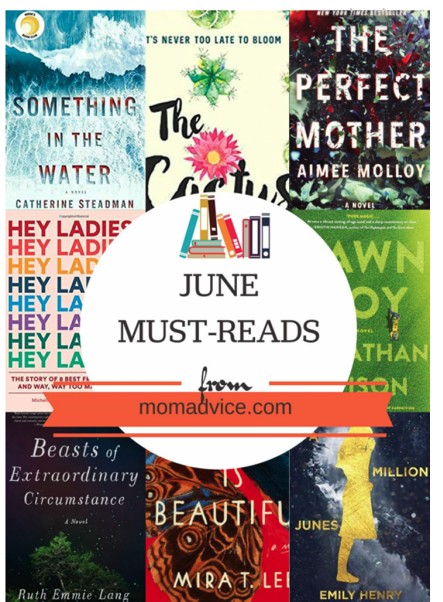

```
Given OCR-based Page Parser Results:        THE T'S NEVERTOO LATE TO BLOOM
                      PERFECT
 SOMETHING
       The
                MOTHER INTHE
     WATER          AIMEEMOLLOY
     NOVE
 CATHERINE STEADMAN
 HEY LADIE
 HEY LADI'
                WN
 HEY LAD
          JUNE HEY LA
 HEY LA
 HEY LA    MUST-READS
 HEY LA
              THAN HEY LA
 THE STORYOF8 BEST FA
 AND WAY,WAYTOOM
        momadvice.com
  Beasts of
  Extraordinary          MILLION
 Circumstance
      BEAUTIFL
  A Novel
            JUNES
         MIRAT.LE
 Ruth Emmie Lang      EMILYHENRY
            SOvOTHATplit THEllorl
```

Q: What are the book recommendations from momadvice.com for June, according to the image?

A: The image recommends several books for June, including "Something in the Water", "The Cactus", "The Perfect Mother", "Hey Ladies", "Lawn Boy", "Beasts of Extraordinary Circumstance", "It's Beautiful", and "Junes Million".

Figure 12: A Layout-aware Finetuning Example of Book Cover.

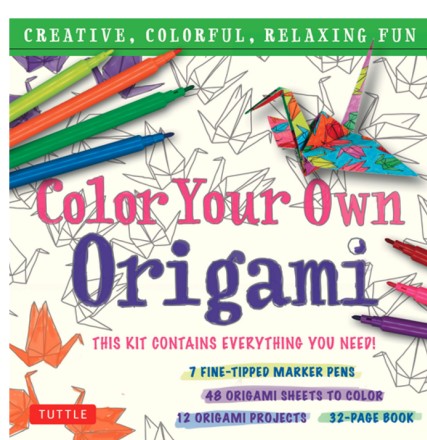

```
Given OCR-based Page Parser Results:
  CREATIVE, COLORFUL, RELAXINGFUN
  ColorYour            Own
   Origami
     THIS KIT CONTAINS EVERYTHING YOU NEED!
          7 FINE-TIPPED MARKER PENS
           48 ORIGAMI SHEETS TO COLOR
  TUTTLE      12 ORIGAMI PROJECTS    32-PAGE BOOK
```

Q: What is the main activity this kit is designed for?
A: This kit is designed for coloring and folding origami.

Figure 13: A Layout-aware Finetuning Example of Book Cover.

| No. | User Instruction |
|---|---|
| 1 | Given the OCR results, could you recover the layout information in the image and reorganize the texts? |
| 2 | Using the OCR results, can you retrieve the layout information from the image and rearrange the texts? |
| 3 | Can you utilize the OCR results to extract the image's layout information and restructure the texts? |
| 4 | Given the OCR results, would you be able to reconstruct the layout of the image and reorganize the text? |
| 5 | Could you use the OCR results to recover the layout details from the image and then rearrange the text? |
| 6 | Based on the OCR results, can you restore the layout information in the image and reposition the texts? |
| 7 | With the OCR results, could you recapture the image's layout information and reorder the texts? |
| 8 | Using the OCR data, can you regain the layout information from the image and reshuffle the text? |
| 9 | Can you interpret the OCR results to retrieve the layout information of the image and reorganize the text accordingly? |
| 10 | Could you use the OCR findings to recover the image's layout information and restructure the texts? |

Table 8: Task III: Text Layout Reconstruction Templates

| No. | Request |
|---|---|
| 1 | Could you extract the layout details from the image provided and rearrange the text accordingly? |
| 2 | Please analyze the image's structure and reformat the text based on its layout. |
| 3 | Can you decipher the layout of the image and restructure the text elements as they appear? |
| 4 | I need you to interpret the layout information within the image and reposition the texts to mirror that layout. |
| 5 | Would you be able to delineate the layout from the given image and reorder the text content accordingly? |
| 6 | I request that you retrieve the spatial arrangement of the image and reconfigure the text to align with it. |
| 7 | Please deduce the compositional layout of the image and systematically reassemble the text. |
| 8 | Can you outline the image layout and reconstruct the text placements to correspond with it? |
| 9 | I'm looking for an analysis of the image's layout so you can reorganize the text segments based on their original positioning. |
| 10 | Kindly dissect the layout patterns in the image and resequence the text in harmony with those patterns. |

Table 9: Task IV: Page Parser Templates

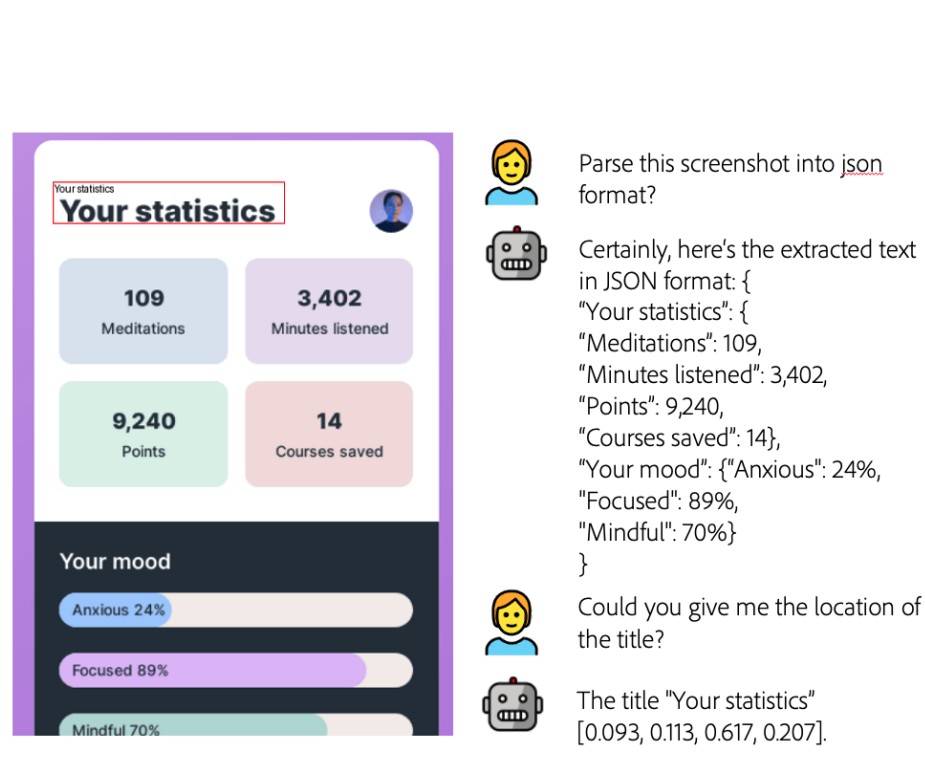

Figure 14: A generated example of text-grounding on screenshot.

