# OpenReview forum: "LLaVA-Read: Enhancing Reading Ability of Multimodal Large Language Models"
_ICLR.cc/2025/Conference — Submitted to ICLR 2025_

### Official Review · Reviewer_ac74 · 2024-10-29

**Soundness:** 2
**Presentation:** 3
**Contribution:** 2
**Rating:** 3
**Confidence:** 5

**Summary:**

The paper presents LLaVA-Read, which utilizes dual visual encoders along with a visual text encoder. The model surpasses existing mllm models in various text-rich image understanding tasks.

**Strengths:**

1. The method is relatively simple and easy to reproduce；
2. The experiment verified the effectiveness of the method and conducted some ablation studies.

**Weaknesses:**

1. The innovativeness of the method is limited, as it is based on multiple visual encoder works, such as Mini-Gemini, mPLUG-DocOwl2, and others.
2. The method of integrating OCR is not very flexible, and the lack of OCR information can affect MLLM, for example, the accuracy and quantity of OCR.

**Questions:**

see weaknesses.

---

### Official Review · Reviewer_q5FL · 2024-11-02

**Soundness:** 3
**Presentation:** 3
**Contribution:** 3
**Rating:** 6
**Confidence:** 3

**Summary:**

The paper discusses the limitations of existing multimodal large language models in understanding textual content within images, primarily due to inadequate text recognition and layout comprehension. To address these issues, the authors introduce LLaVA-Read, a multimodal model that employs dual visual encoders alongside a dedicated visual text encoder. Their model outperforms current state-of-the-art models in tasks that require understanding text-rich images. The research highlights that effective visual text understanding is a significant challenge and emphasizes the importance of a robust visual text encoder for future multimodal systems.

**Strengths:**

The paper analyzes the text recognition capabilities of multimodal large language models, which reveals their impressive capability on scene text understanding but limited proficiency in comprehending large amounts of textual content within a text-rich image.

The proposed LLaVA-Read surpassing multiple baselines of text-rich image understanding

**Weaknesses:**

It seems that the model's performance improvement largely stems from carefully designed pre-training tasks. How does the model's design regarding high and low resolutions aid in text understanding? What is the authors' perspective on this issue?


Some similar works, such as Ferret UI [a] and CogAgent [b], adopt similar ideas. Could the authors discuss the differences and advantages compared to them?

[a] You et al. https://arxiv.org/pdf/2404.05719
[b] Hong et al. https://arxiv.org/pdf/2312.08914

**Questions:**

Please refer the weakness.

---

### Official Review · Reviewer_kYAq · 2024-11-03

**Soundness:** 2
**Presentation:** 1
**Contribution:** 1
**Rating:** 3
**Confidence:** 4

**Summary:**

This study combines advanced techniques from prior studies on text-rich image understanding to create a more effective multimodal large language model (MLLM). Specifically, it uses multiple vision encoders, a fusion method for visual tokens, and auxiliary information from an off-the-shelf OCR tool. Additionally, it incorporates layout recovery methods, enhancing the model’s ability to capture complex text and visual layouts within images. This approach allows the proposed model, LLaVA-Read, to achieve comparable performance compared to existing models in text-rich image tasks by better handling visual and textual elements in images.

**Strengths:**

1. This study successfully integrates several recent techniques proposed in multimodal large language models (MLLMs), particularly those with strengths in text-rich image understanding, including the following:
- The introduction of a dual encoder system to effectively handle both high- and low-resolution image pathways and a fusion method to compress the length of vision tokens efficiently (as demonstrated in LLaVA-HR by Luo et al., 2024).
- An off-the-shelf OCR tool, PaddleOCR, is used as an auxiliary encoder to enhance text recognition (from the approach used in LLaVAR by Zhang et al., 2023d).
- The implementation of a layout recovery process to better utilize layout information is inspired by the LATIN-Prompt method (Wang et al., 2023a).

2. The study validates the proposed model's performance through various experiments, leading to valuable findings. Notably, findings 1-3 in the paper could significantly inform future research. The comparison in Figure 2 between a commercial OCR tool and CLIP, a widely used vision encoder in the field, across different font sizes is especially noteworthy, offering insights into recognition performance under various conditions.

**Weaknesses:**

1. Explanation of Methodological Differences

- The paper could benefit from a more precise explanation of how the proposed methods differ from existing approaches. For instance, the Mixed Resolution Adaptor (MRA) appears similar to LLaVA-HR but has certain distinct elements. While previous methods fuse the final three layers, the proposed approach fuses the 1st, 4th, and 7th layers, with slight differences in notation. Additionally, it is still being determined whether the layout recovery process used in the OCR Tokenizer is identical to the existing method. Although sharing source code could clarify these issues, providing detailed explanations within the manuscript or appendix would help readers understand the unique contributions of this study and avoid potential novelty issues.

2. Benchmark and Performance Reproduction

Given that this study integrates multiple techniques from previous work, confirming their synergy through benchmarks is crucial. However, there are some concerns regarding performance evaluation:
- The main performance results in Table 1 differ from the values reported in prior studies, with some benchmarked models performing lower than expected. This discrepancy calls for a more transparent explanation of the reproduction process. For example, DocVQA scores for EAGLE (Shi et al., 2024) are reported as 86.6 (Llama3-8b) and 85.4 (Vicuna-13B), but in this study, the reproduced score is 68.5, indicating a significant gap. Similar trends are observed with Idefics-2 and others. Although exact reproduction is challenging due to differences in model architecture, pretraining and fine-tuning data, and training methods, a clear explanation is more significant since the study emphasizes the high performance of the proposed model.
- The table should clarify model names for the alternatives. To ensure fair comparisons, models with smaller parameters than the proposed approach's Vicuna-13B should be labeled accordingly (e.g., Idefics-2 at 8B and EAGLE at 8B and 13B).

3. Coverage of Related Research

The paper would benefit from including additional key studies in related work:
- The use of external OCR tools, as employed in CREAM (Kim et al., 2023).
Multiple Vision Encoders have been used in models like the SPHINX series (Lin et al., 2023) and DeepSeek-VL (Lu et al., 2024).
- Essential works in text-rich document understanding, such as UDOP (Tang et al., 2023) and Idefics3 (Hugo et al., 2024), should also be referenced.
- Although recent papers like Llama-3.2 (META, 2024) and Qwen2-VL (Peng et al., 2024) may not have been available during submission, they have achieved notable performance metrics and could be referenced in future work.

Including these studies in the Related Work section would enrich the paper's context.


4. Ablation Study Clarifications

There are some areas where the ablation study could be improved:
- The apparent effect of layout recovery has yet to be fully demonstrated. A baseline comparison with methods like layoutLM or pix2struct would be beneficial. If the LATIN-Prompt (Wang et al., 2023a) method is used exactly as in prior work, a mention of its verified effectiveness in those studies will clarify this.
- Providing additional performance metrics for ablation would help, as the current Table 3(b) lacks "recognition score" and "total". Including these and more tasks from Table 1 beyond OCRBench would strengthen the ablation analysis.
- The ablation studies should ideally fix the proposed model's resolution. Since most ablation studies in Table 3 are conducted at 1024 resolution, this may be the default resolution. Also, confirming whether the LLaVA-Read results in Table 1 are all based on 1024 resolution would add clarity.
- Lastly, if the two vision encoders differ from those in LLaVA-HR, explaining the criteria used for selecting them would enhance the transparency of the model's design choices.

**Questions:**

1. The notation on P4L213 appears different from the one on P5L217, and it would be helpful to clarify how this diverges from the Mixed-Resolution Adaptation (MRA) approach initially proposed in LLaVA-HR.

2. In P7L375, "LLaVA-Read-H" is mentioned. Does this refer to the high-resolution encoder among the dual encoders? Does this imply two versions, LLaVA-Read (768x768) and LLaVA-Read-H (1024x1024)? Clear definitions and consistent mention of these versions across results would be beneficial.

3. Are the number of vision tokens constrained by the low-resolution encoder, resulting in 576 (24 x 24) tokens?

4. The study denotes performance improvement when the resolution increased from 768 to 1024. Would further increasing resolution continue to yield benefits? Are there additional costs associated with higher resolution?

5. Is the image in Table 2 randomly selected? It might be more accurate to replace it with an example highlighting specific errors, as the current selection could lead to misunderstandings regarding GPT-4V's performance. Notably, GPT-4V achieves the highest performance in the KIE benchmark. For instance, is the expectation in Table 2 that "BRN" should be recognized as "Brown"? Clearly distinguishing between misrecognition and phrasing differences could help clarify results.

6. If basic tokenization was applied in the OCR tokenizer's $f_q$ step, consecutive whitespaces could have been removed, potentially diminishing the layout recovery process's value. Confirming whether a specialized tokenizer was used to address this would be helpful.

7. Please provide details on the token length distribution of the OCR tokenizer results. OCR information can significantly benefit scene-text or document image processing but may consume a substantial part of the LLM's context capacity. Comparing the OCR token lengths generated by the proposed OCR tokenization with those from other methods could offer additional insights.

---

### Meta-Review · Area_Chair_G3KF · 2024-12-19

**Metareview:**

The reviewers raised two main concerns about (1) insufficient conceptual and experimental comparisons with similar approaches and (2) concerns regarding presentation clarity. However, the authors did not address any of these questions during the rebuttal period. Since these important issues remain unresolved, I recommend rejection for this paper.

For future submissions, I recommend:
- Adding thorough comparisons with similar approaches
- Improving the manuscript organization and presentations

These improvements would help better demonstrate the paper's contributions.

**Additional Comments On Reviewer Discussion:**

The authors did not respond during the rebuttal period.

---

### Decision · Program_Chairs · 2025-01-22

Reject